# Possible Potentiating Effects of Combined Administration of Alcohol, Caffeine, and Nicotine on In Vivo Dopamine Release in Addiction-Related Circuits Within the CNS of Rats

**DOI:** 10.3390/biomedicines12112591

**Published:** 2024-11-13

**Authors:** Carmen Costas-Ferreira, Martiño Barreiro-Chapela, Rafael Durán, Lilian R. Ferreira Faro

**Affiliations:** Department of Functional Biology and Health Sciences, Faculty of Biology, University of Vigo, Campus As Lagoas-Marcosende, 36310 Vigo, Spain; maica.cf@hotmail.com (C.C.-F.); martinobarreiro@gmail.com (M.B.-C.); rduran@uvigo.es (R.D.)

**Keywords:** dopamine and drugs, signaling pathways, neuromodulation, psychoactive substances, microdialysis, rat

## Abstract

Background: Studies that assess the effects of the interaction of psychoactive substances on dopamine release, the key neurotransmitter in the neurochemical and behavioral effects related to drug consumption, are crucial to understand both their roles and the dysfunctions they produce in the central nervous system. Objective: We evaluated the effects of individual and combined administration of the three most widely consumed psychoactive substances in the world, ethanol, caffeine, and nicotine, on dopaminergic neurotransmission in three brain regions of rats related to addiction: the prefrontal cortex (PFC), the nucleus accumbens (NAcc), and the dorsal striatum. Methods: The dopamine levels were measured in vivo by cerebral microdialysis associated with HPLC-ED. Results: We observed that local administration of a single concentration of caffeine (5 mM) or nicotine (5 mM) significantly increased the dopamine levels in all three areas studied, while ethanol (300 mM) increased them in the NAcc and striatum. Perfusion of nicotine + caffeine produced a synergistic effect in both the NAcc and striatum, with increases in the in vivo dopamine release greater than the sum of the effects of both substances. When administering the combination of nicotine + caffeine + ethanol, we observed an additive effect in the NAcc, while in the PFC we observed a synergistic effect. Conclusions: Our results support the stimulating effects of caffeine, nicotine, and ethanol on the brain reward system. In addition, we also observed that the administration of different mixtures of these substances produces synergistic and additive effects on the release of dopamine in the mesocortical and nigrostriatal systems.

## 1. Introduction

The term “polydrug use” refers to the combined consumption of two or more drugs over a defined period, either simultaneously or at different times, for therapeutic or recreational purposes [1]. Generally, the concurrent use of psychoactive substances is done to experience new effects not achievable with the consumption of individual substances, enhance desired effects, or counteract the negative effects caused by the consumption of other substances [2]. In all cases, the simultaneous consumption of psychoactive substances increases the physical and emotional risks for the individual, raising the likelihood of developing mental or behavioral disorders or cognitive impairment, among other effects [1,3,4,5].

Polydrug use or multiple substance use generally refers to the consumption of multiple illicit drugs, but can also include legal substances or medications, such as the combination of alcohol and nicotine or the consumption of alcohol with anxiolytics or stimulants. In general, combined use of psychoactive substances typically involves a primary substance and one or more secondary substances, as seen in the case of alcohol with caffeine or tobacco, or alcohol with anxiolytics. While many effects of these three substances are well-known, little is understood about the neurochemical effects produced by their interaction in the central nervous system (CNS). This is because studies evaluating the effects of combined exposure to these substances describe their effects after chronic, systemic administrations, typically involving only two substances and rarely three or more. Therefore, studies that assess the effects of the interaction of psychoactive substances on dopamine release, the key neurotransmitter in the neurochemical and behavioral effects related to drug consumption, are crucial to understand both their roles in this system and the dysfunctions they produce.

The development and maintenance of addictive behaviors result from maladaptive dysregulations within the neural circuits responsible for decision-making, learning, motivation, and reward processing [6]. Some components of these circuits include (1) the nucleus accumbens (NAcc), which plays a fundamental role in mediating the reinforcing effects of drugs; (2) the prefrontal cortex (PFC), which constitutes a key region for reward, behavioral control, and flexibility; and (3) the basal ganglia, which mediates substance-seeking and consumption behaviors [7,8,9].

Nicotine, caffeine, and ethanol are substances whose consumption is legally permitted. They are the three most consumed psychoactive substances in the world and are frequently consumed together [10,11]. The common neurochemical mechanism of action for these substances, and all drugs of abuse, is the increase in dopamine levels in the mesocorticolimbic pathway [12,13]. This dopaminergic circuit consists of cell bodies located in the ventral tegmental area (VTA), which send ascending projections rostrally to limbic regions such as the NAcc and the PFC. Their activation is associated with reward-related learning and motivated behaviors, among other functions [14,15]. Furthermore, the striatum is particularly involved in the later stages of addiction, where there appears to be a neuroanatomical progression from the ventral striatum (NAcc) to the dorsal striatum, where drug-seeking and drug consumption behavior originate [16,17].

Considering the relevance of the NAcc, the PFC, and the dorsal striatum in the development and maintenance of addictions, in the present study, we have evaluated the effects of caffeine, nicotine, and ethanol administered individually and in different combinations on the in vivo dopamine release in the rat mesocorticolimbic and nigrostriatal pathways, specifically in the NAcc, the PFC, and the striatum. Our main objective was to evaluate how the mixture of these substances can affect dopaminergic neurotransmission in the CNS. This is because, depending on how they influence each other, different types of interactions can occur: additive effects, when the total effect of two or more substances is equal to the sum of their individual effects; synergistic effects, where the combined effect of the substances is greater than the sum of their individual effects; potentiating effects, in which one substance alone has no toxic effects but, when combined with another, increases the toxic effect of the latter; and antagonistic effects, which occur when the combination of two substances reduces their expected toxic effect. Our results show that different combinations of nicotine, caffeine, and ethanol produce synergistic, additive, and potentiating effects on the in vivo dopamine release.

## 2. Methods

### 2.1. Animals and Treatments

For the experiments, adult female Sprague-Dawley rats (250–350 g) were used. These animals were specifically bred for experimental purposes and provided by the Breeding Facility of the CINBIO (*Centro de Investigaciones Biomédicas*) of the University of Vigo (Spain). The animals were housed in polypropylene cages with a top grid and a food receptacle (dimensions: 425 mm × 266 mm × 185 mm), under constant humidity (45–65%) and temperature conditions (22 ± 2 °C), with controlled light and dark periods (14 and 10 h, respectively), until the day of the experiment. The rats were fed with commercial feed and tap water, which were available ad libitum.

All experiments were conducted following the European Union Council (2010/63/EU) and the current legislation of the Spanish State (*Real Decreto* 53/2013) regarding the use of animals for experimental purposes. The study was approved by the Ethics Committee on Animal Welfare of the University of Vigo and Xunta de Galicia (ES360570215601/21/FUN01/BIOL.AN.08/LFF/01). Every effort was made to minimize the suffering and distress of the animals.

Caffeine, nicotine, dopamine, 3,4-dihydroxyphenylacetic acid (DOPAC), homovanillic acid (HVA), and glacial acetic acid were acquired from Sigma-Aldrich (St. Louis, MO, USA). Ethanol (99.8% purity) was obtained from PRS PanReac (Barcelona, Spain). All reagents and substances used were of analytical grade. The water used for preparing standard solutions, reagents, and chromatographic solvents was obtained from a Milli Q ultrapure water system (Millipore, Billerica, MA, USA).

Caffeine, nicotine, and ethanol were diluted in the perfusion medium (Ringer) to the required concentrations and locally administered in the different areas studied. The concentrations used (5 mM nicotine and caffeine and 300 mM ethanol) were chosen based on data from the literature where similar concentrations were used for this type of study [18,19,20,21,22].

In the present study, we evaluated the effects of the different substances exclusively in female rats. This choice is based on relatively recent studies indicating that there are no appreciable differences between sexes in neurochemical, electrophysiological, histological or behavioral measures [23,24]. Likewise, in some of the previous studies carried out in our laboratory, no significant differences were found between males and females regarding basal dopamine levels and the effect produced by the administration of different treatments [25,26,27].

### 2.2. Microdialysis Procedure

The microdialysis technique was performed according to previous studies conducted in our laboratory [26,27,28]. Briefly, after anesthetizing the animal with chloral hydrate (400 mg/kg i.p.), it was placed in a stereotaxic apparatus (Nashigire SR-6, Kyoto, Japan), and a small incision was made in the head skin to expose the skull surface. Afterward, a small hole was drilled in the skull surface, and a CMA12 guide cannula (CMA/Microdialysis, Solna, Sweden) was implanted in the PFC, NAcc, or striatum, according to the coordinates established by Paxinos and Watson [29], with Bregma as a reference point. Since three different brain areas were studied in this work, the coordinates used to implant the cannula in each of them were different. Figure 1 shows schematic diagrams illustrating the implantation points of the microdialysis probes in the three different regions studied.

Twenty-four hours after the surgical procedure, a CMA12 microdialysis probe (0.5 mm diameter, 2 mm membrane length for the PFC and NAcc, or 3 mm membrane length for the striatum) (CMA/Microdialysis, Solna, Sweden) was inserted through the guide cannula. The probe was connected to a CMA/402 infusion pump (CMA/Microdialysis, Solna, Sweden) and perfused with a Ringer’s solution (147 mM NaCl, 4 mM KCl, and 2.4 mM CaCl_2_, pH = 7.4) at a constant flow rate of 1.5 μL/min. Experiments were conducted with awake, conscious, and freely moving animals.

### 2.3. Experimental Design

All experiments followed the same general protocol and had a total duration of 3 h, involving a total of 81 animals divided into 15 experimental groups. The experimental groups are shown in Table 1. After a period for stabilization, the experiments started with Ringer’s solution perfusion during the first 60 min to collect three baseline samples. Subsequently, the treatment was administered during the second hour of the experiment (60 min, three samples). In the last hour of the experiment (60 min), Ringer’s solution was perfused again to collect another three samples under initial conditions (Table 1). The administration of treatments was performed locally through a microdialysis probe in the PFC, NAcc, or striatum.

At the end of the microdialysis experiments, the animals were deeply anesthetized and rapidly euthanized by cervical dislocation. A craniotomy was performed for histological confirmation of the correct probe placement. Only data from animals with correct placements of the probe in the PFC, NAcc, or striatum were used. Seven animals were found with incorrect location of the microdialysis probe and were not used in the study.

### 2.4. HPLC Conditions

Dopamine and its acidic metabolites’ (DOPAC and HVA) levels in the microdialysis samples were quantified using high-performance liquid chromatography with electrochemical detection (HPLC-ED) following the protocol developed by Durán et al. [30] with slight modifications. Dialysate samples (30 μL) were collected every 20 min using a CMA/142 microfraction collector (CMA/Microdialysis, Solna, Sweden) and immediately injected into an HPLC system using a Rheodyne 7125 injection valve. Isocratic separation of dopamine, DOPAC, and HVA was achieved using Dionex C18 reverse-phase columns (particle size 5 μm). The mobile phase consisted of 70 mM KH_2_PO_4_, 1 mM octanesulfonic acid, 1 mM EDTA, and 14% methanol (pH 3.4). Elution was performed at a flow rate of 1.0 mL/min using a Jasco PU 1580 pump (Jasco, Tokyo, Japan). Substance detection was achieved using an ESA Coulochem III 5100A electrochemical detector (ESA, Chelmsford, MA, USA) with an oxidation potential of +400 mV. Data were analyzed using the Chromanec XP 1.0.4 chromatographic software (Micronec, Barcelona, Spain).

### 2.5. Data Analysis

All data were corrected using the percentage of in vitro recovery for every microdialysis probe, which was similar for the different types of probes used and for the substances analyzed: 12.5% for dopamine, 20.1% for DOPAC, and 23.2% for HVA.

Data are presented as mean ± S.E.M. of 5 to 7 animals in each group. Baseline levels of dopamine, DOPAC, and HVA (defined as 100%) were determined from the two samples before treatment administration. Results were calculated as percentages of baseline levels. Data on dopamine and its metabolites were corrected using the in vitro recovery percentage for each microdialysis probe.

Statistical analysis of the results was performed employing ANOVA and Scheffe *post-hoc* test, considering the following significant differences: *p* < 0.05, *p* < 0.01, and *p* < 0.001.

## 3. Results

Periodical control experiments have been carried out under our microdialysis conditions to confirm the basal values and the adequacy of our conditions. Basal levels of dopamine and its metabolites in dialyzed samples were stable in control animals (non-treated rats) and before the administration of treatment. The mean of dopamine, DOPAC, and HVA concentrations in the two samples collected before caffeine, nicotine, ethanol, or their mixtures administration was considered as the basal levels: 0.048 ± 0.004, 1.71 ± 0.08, and 0.19 ± 0.02 ng/μL, respectively.

Intrastriatal administration of caffeine, nicotine, ethanol, or their combinations at the tested concentrations did not produce seizures, tremors, or other types of apparent behavioral or physiological dysfunctions.

### 3.1. Effects of Single Administration of Caffeine, Nicotine, or Ethanol on Dopamine, DOPAC, and HVA Levels

The infusion of 5 mM caffeine (60 min) significantly increased extracellular dopamine levels in the NAcc, PFC, and striatum up to 437.6 ± 39.1% (*p* < 0.001), 448 ± 16.6% (*p* < 0.001), and 461.8 ± 74.1% (*p* < 0.01) relative to baseline levels, respectively (Figure 2A). This maximum increase was observed 20 min after the start of treatment in the striatum, but it was recorded later (at 40 min) in the NAcc and PFC. The effect of caffeine on dopamine levels in the striatum at 40 min was significantly lower than that observed in the NAcc (*p* < 0.05) and PFC (*p* < 0.01). Additionally, while baseline dopamine levels fully recovered in the striatum and NAcc in the last 60 min of the experiment, in the PFC, they remained elevated until the end, not returning to baseline values.

Local administration of nicotine (5 mM, for 60 min) induced significant increases in dopamine levels in the NAcc, PFC, and striatum, reaching maximum values of 249 ± 41.7% (*p* < 0.001), 581.7 ± 30% (*p* < 0.001), and 311.8 ± 48.1% (*p* < 0.001), compared to baseline levels, respectively (Figure 2B). As observed for caffeine, this maximum increase was seen at 20 min in the striatum, while in the NAcc and PFC, the maximum effect was recorded at 40 min. The effect of nicotine on dopamine release was significantly greater in the PFC than in the NAcc or striatum (*p* < 0.01). Additionally, while baseline dopamine levels fully recovered in the NAcc and striatum in the last 40 min of the experiment, in the PFC, they remained elevated until the end, not returning to baseline values.

When ethanol (300 mM, for 60 min) was administered in the studied brain areas, dopamine concentrations significantly increased in the NAcc and striatum to 284.7 ± 50.3% (*p* < 0.001) and 269.3 ± 67.2% (*p* < 0.001), compared to baseline values, respectively. The ethanol administration had no significant effect on dopamine levels in the PFC. The maximum increase occurred 40 min after the start of ethanol administration, and baseline dopamine levels recovered by the end of the experiment in both areas (Figure 2C). As shown in Figure 2C, the greatest effect of ethanol was observed in the striatum and NAcc, and this effect was significantly greater than that observed in the PFC (*p* < 0.05).

Regarding dopamine metabolites, the results show considerable variability in the different areas studied. In the NAcc, significant increases in DOPAC levels were observed after the administration of caffeine or ethanol. In the PFC, DOPAC levels significantly increased after caffeine or nicotine administration. In the striatum, only caffeine and ethanol induced significant increases in the levels of this metabolite (Table 2). Concerning HVA levels, the results show that nicotine increased its levels in the NAcc, whereas none of the administered treatments produced statistically significant changes in HVA levels in the PFC. In the striatum, caffeine, nicotine, and ethanol significantly increased the HVA levels (Table 2).

### 3.2. Effects of Administration of Different Combinations of Psychoactive Substances on Dopaminergic Neurotransmission

#### 3.2.1. Effects of Co-Administration of Caffeine + Nicotine or Caffeine + Nicotine + Ethanol on Dopamine Levels in the NAcc

The effects of the administration of different combinations of psychoactive substances in the NAcc are shown in Figure 3. The co-administration of nicotine (5 mM) with caffeine (5 mM) significantly increased dopamine levels, reaching a peak of 1036.2 ± 160.3% (*p* < 0.001), regarding the baseline, 40 min after the start of mixture administration (Figure 3A). The effect of coadministration was statistically greater than the effect observed with nicotine (*p* < 0.01) or caffeine alone (*p* < 0.05).

Figure 3B shows that the perfusion of the mixture of nicotine, caffeine, and ethanol in the NAcc increased dopamine levels to 904.4 ± 193.1% (*p* < 0.01), compared to baseline values, 40 min from the start of treatment. When comparing this effect with those obtained with individual administrations, it is observed that it was significantly lower than the one induced by the perfusion of nicotine, caffeine, or ethanol alone (*p* < 0.05).

#### 3.2.2. Effects of Co-Administration of Caffeine + Nicotine or Caffeine + Nicotine + Ethanol on Dopamine Levels in the PFC

The administration of the nicotine and caffeine mixture significantly increased extracellular dopamine levels in the PFC to a peak of 1043.2 ± 95.8% (*p* < 0.001) compared to the baseline levels. This maximum effect was observed 60 min after the start of the co-administration of the substances, and dopamine levels returned to baseline values at the end of the experiments (Figure 4A). This effect was statistically greater than the effect observed with caffeine alone (*p* < 0.05).

In Figure 4B, the effects of intracortical perfusion of nicotine, caffeine, and ethanol combination on dopamine levels in the PFC are shown and compared with the effects induced by these substances individually. The results demonstrate that the co-administration of the mixture induced significant increases in dopamine levels, reaching a peak of 1521 ± 259.6% (*p* < 0.001) compared to baseline levels. This maximum increase was observed at 40 min from the start of the co-administration, and extracellular dopamine levels returned to baseline values in the last 20 min of the experiment. The effect produced by the perfusion of the three substances was significantly greater than that produced by the individual administration of nicotine (*p* < 0.01), caffeine (*p* < 0.01), or ethanol (*p* < 0.01).

#### 3.2.3. Effects of Co-Administration of Caffeine + Nicotine or Caffeine + Nicotine + Ethanol on Dopamine Levels in the Dorsal Striatum

The co-administration of nicotine and caffeine in the striatum induced a summative effect on dopamine levels in this area. Specifically, the perfusion of nicotine and caffeine increased dopamine release to 918.3 ± 96.4% (*p* < 0.001) compared to baseline values 40 min from the start of the administration of the substances (Figure 5A). The effect produced by the co-administration of both substances on in vivo dopamine release from the striatum was greater than that recorded after the individual infusion of nicotine (*p* < 0.05) or caffeine (*p* < 0.01).

On the other hand, in Figure 5B, it can be observed that the administration of the mixture of nicotine, caffeine, and ethanol produced a maximum effect on striatal dopamine levels [311.2 ± 76.5% (*p* < 0.001)] at 40 min from the start of the perfusion, which was significantly lower than that observed after ethanol perfusion alone (*p* < 0.05).

#### 3.2.4. Effects of Co-Administration of Caffeine + Nicotine or Caffeine + Nicotine + Ethanol on DOPAC and HVA Levels

The effects of the administration of different mixtures on the levels of dopamine metabolites are shown in Table 2. As observed for individual administration, the treatment of animals with different mixtures produced diverse effects on both DOPAC and HVA levels. The additive effect of the mixture of nicotine, caffeine, and ethanol on DOPAC levels in the PFC is noteworthy.

Figure 6 shows a comparison of the maximal effects observed after the administration of nicotine, caffeine, or ethanol, and the combinations of nicotine + caffeine and nicotine + caffeine + ethanol on dopamine levels in the NAcc, PFC, and striatum. The graphs show the synergistic effect of the administration of caffeine together with nicotine in the NAcc and striatum, and that the addition of ethanol to this mixture produces an additive effect on the release of dopamine.

## 4. Discussion

In the present research, we observed that administration of a single concentration/dose of the psychoactive substances studied, either individually or in combination, produced a series of effects on extracellular dopamine levels that varied across different areas studied and the combination used. Under our experimental conditions, in situ administration through a microdialysis probe of nicotine (5 mM), caffeine (5 mM), or ethanol (300 mM), in most cases, resulted in significant increases in extracellular levels of dopamine and its metabolites in the NAcc, PFC, and striatum of rats. Our results also show that administration of different combinations of the substances stimulated the dopamine overflow, and in several cases, the release was significantly greater than that observed with individual administrations. When studying the effects of mixtures of substances, the main objective is to determine whether the observed effect follows a simple additive approach or deviates to generate a synergistic, antagonistic, or potentiating effect. The chemical composition of the mixture and the mechanism of action of the individual substances are the factors that will determine the result observed. The greater effect of the combinations of psychoactive substances on dopamine release demonstrates, from a neurochemical standpoint, the potential additive, synergistic, and antagonist effects of the coadministration of two or more psychoactive substances.

### 4.1. Effects of Caffeine and Nicotine Combination

Several studies indicate that exposition of experimental animals with different doses, exposure times, and modes of administration of caffeine produces stimulant effects on dopaminergic neurotransmission in the mesocorticolimbic and nigrostriatal pathways [31,32,33,34,35]. However, the different research groups observed disparities both concerning the areas studied and the profile of the dopamine increases. In the present study, we observed that the in situ administration of a single concentration of caffeine produced significant increases in extracellular levels of dopamine and its metabolites DOPAC and HVA in the three regions studied.

Although the present study did not assess the mechanism of action by which caffeine increases dopamine release, substantial literature suggests that this substance exerts its effects through an antagonistic action on adenosine receptors A1 and A2A (A1R and A2AR) [36,37]. However, available data also show that acutely administered caffeine primarily exerts its effects by acting on presynaptic A1Rs, which are abundant in the mesocorticolimbic and nigrostriatal pathways [38,39]. Therefore, based on these findings, we hypothesize that the increases in the in vivo dopamine release observed in our study are mainly due to the antagonistic action of caffeine on A1Rs located in dopaminergic terminals in the three regions, consistent with effects observed by other authors [21,40,41]. Additionally, A1R is also present in glutamatergic terminals in the NAcc and striatum, and its inhibition by caffeine increases both glutamate and dopamine release in these regions. It has been demonstrated that, in this case, dopamine release occurs indirectly and is dependent on the elevation of glutamate levels and activation of its ionotropic receptors in dopaminergic terminals [42,43].

Like what has been observed for caffeine, the administration of nicotine also produced significant increases in the in vivo release of dopamine and its metabolites in the three areas studied. The stimulatory effect of nicotine on in vivo dopamine release was likely due to its action on nicotinic acetylcholine receptors (nAChR), which are abundantly present in the mesocorticolimbic and nigrostriatal pathways [44,45,46,47]. It has been demonstrated that, to induce dopamine release, nicotine primarily activates high-affinity nAChRs (α4β2, α6β2, or α4α6β2) present in dopaminergic terminals [48,49]. Generally, in the three regions studied, this release depends almost exclusively on nAChRs that possess β2 subunits and are located presynaptically [44,48,50,51]. Nevertheless, nicotine also appears to act on α7-type nAChRs located in glutamatergic terminals in the mesolimbic and nigrostriatal pathways, whose activation enhances glutamate release and, consequently, indirect dopamine release [45,52].

A very common social behavior among people is to associate the consumption of caffeine (coffee) with nicotine (smoking a cigarette), and each of them could reinforce the action of the other. This reinforcement has been demonstrated from a neurochemical point of view in the present study, where we observed that nicotine plus caffeine administration significantly increased dopamine overflow in NAcc, PFC, and striatum (NAcc: 1036%; PFC: 1043%; striatum: 918%), these increases being significantly greater than those observed after individual nicotine or caffeine administration. These results also highlight that the combination of nicotine and caffeine possibly produces a synergistic type of interaction in both the NAcc and striatum but not in the PFC, where we observed that the total effect on dopamine release was approximately comparable to the sum of the total increase induced by the substances individually (NIC: 582%; CAF: 448%; NIC + CAF: 1043%) (Figure 6).

Although in the present study we did not evaluate the possible mechanisms underlying these observed effects, we hypothesize that nicotine may induce dopamine release through the activation of nAChRs with β2 subunits and indirectly through α7-type nAChRs present in glutamatergic terminals. On the other hand, caffeine, by blocking mainly presynaptic A1Rs, would suppress the inhibition imposed by the activity of these receptors on dopamine release, thus increasing its extracellular levels (Figure 7A). However, our results show that the nicotine and caffeine mixture increases dopamine levels beyond these individual mechanisms (a non-interaction). The synergistic effect observed by us can indicate that the interaction between the two substances is more complex and is not explained by a simple sum of effects. Further research is needed to determine what kind of interactions occur between the two substances to induce dopamine release in the mesolimbic pathway.

In the available literature, the only previous study investigating the effects of the co-administration of nicotine and caffeine on dopaminergic neurotransmission was the in vitro study conducted by Garção et al. [53] using synaptosomes, where co-administration did not have an additive effect on dopamine release. These authors suggested that striatal dopamine release would be modulated by a presynaptic interaction between A2AR and α6β2 nAChRs but would be independent of α7 nAChR activation. These findings contrast with our results, where the perfusion of caffeine along with nicotine in the NAcc and striatum produced a synergistic effect. The differences between the two studies could lie in the fact that while Garção et al. [53] evaluated [3H]dopamine release in synaptosomes from dopaminergic terminals, our study used a structurally preserved system, and dopamine levels were quantified in a physiologically intact environment.

### 4.2. Effects of Caffeine, Nicotine, and Ethanol Combination

The positive motivational properties of ethanol, responsible for its consumption abuse and alcoholism, depend to some extent on the activation of the mesolimbic dopaminergic system. The activation of dopaminergic neurotransmission in the mesolimbic pathway plays a crucial role in substance dependence in general and alcohol dependence in particular, especially when this system is dysregulated.

In this study, the local administration of a low concentration (300 mM) of ethanol increased dopamine release in the NAcc (285%) and striatum (270%), although it did not alter dopamine levels in the PFC. Our results are consistent with studies where the acute administration of low or moderate doses of ethanol (up to 2.5 g/kg) was associated with an increase in dopamine release and, consequently, an increase in locomotion in experimental animals [20,54,55,56,57]. However, higher and/or chronic doses of ethanol have been linked to a decrease in locomotion, drowsiness, lethargy, and, from a neurochemical perspective, a decrease in dopamine levels [58,59].

On the other hand, we did not observe significant effects of local ethanol administration on dopamine levels in the PFC. The available evidence on the effects of acute administration of moderate doses of ethanol in this region is contradictory, as some studies described a decrease in dopamine release, while others observed an increase or the absence of significant effects [55]. Since the PFC is crucial for behavior control and cognitive flexibility, resolving these discrepancies is important to elucidate the mechanisms of ethanol action and the subsequent neuroadaptations that lead to loss of control over its intake.

Regarding the neurochemical mechanisms of action of ethanol, its effects are complex and could affect dopaminergic neurotransmission in several ways depending on different factors, including the area of study, dosage, and routes of administration, among others. Since, in our experimental conditions, ethanol was administered locally, this implies that the increases in dopamine release observed in the NAcc and the striatum occurred through a presynaptic-level action. Recent research has demonstrated that the inhibition of presynaptic D2 dopamine receptors by specific antagonists reduces ethanol self-administration in the PFC [60]. Therefore, although we did not study the mechanisms of action of ethanol, its effect on dopamine release observed in this study may be mediated by its interaction with D2 dopamine receptors.

We also evaluated the effect of administering ethanol along with caffeine and nicotine, which is one of the most socially used combinations of legal psychoactive substances. To our knowledge, this is the first study to analyze the effect of the combination of the three most used psychoactive substances in the world on the in vivo release of dopamine. Figure 6 shows that the total increase in dopamine release induced by the mixture containing the three substances in the NAcc (904%) was comparable to the sum of the total increase caused by the substances individually (nicotine 249%, caffeine 438%, ethanol 285%), whereas in the PFC (1521%) an apparent synergistic effect was observed after the administration of the three substances at the same time (nicotine 582%, caffeine 448%, ethanol not significant). Furthermore, in the PFC the increase was greater than that observed for the combination of caffeine + nicotine (1043%). Contradictorily, in the striatum, the mixture of the three substances did not produce significant modifications in the dopamine levels when compared to those induced by caffeine, nicotine, or ethanol administered individually (Figure 6). Considering that the nicotine + caffeine infusion caused a 918% increase in dopamine release and that the addition of ethanol to the mixture resulted in an increase of only 311%, the results seem to indicate that ethanol could be responsible for an antagonist effect in this area.

Thus, in our study, we observed that the greatest effect of the mixture of caffeine, nicotine, and ethanol on dopamine release occurred in the PFC and the NAcc, regions that are part of the brain’s reward system, mediating the motor impulsivity induced by psychostimulants and the rewarding and reinforcing properties of these substances, respectively. Regarding the underlying mechanisms of action for the changes observed in these two brain regions, we can hypothesize that the three substances would induce an increase in dopamine levels through different neurochemical mechanisms (Figure 7B): (1) caffeine primarily inhibits presynaptic adrenergic receptors of type A1; (2) nicotine stimulates nAChRs present in dopaminergic terminals; and (3) we assume that ethanol could act on presynaptic D2 dopamine receptors. Considering that the three substances individually increase dopamine levels through independent mechanisms, the observed results would be compatible with the sum of these different mechanisms to induce the increases in dopamine levels measured by brain microdialysis.

One last aspect to consider is the behavior of the acidic metabolites of dopamine. An upward trend in the levels of both metabolites has been observed in almost all experimental conditions studied. In general, the levels of DOPAC and HVA have experienced a significant increase after the administration of the different substances under study, both individually and in combination, in the three evaluated brain regions. These increases in the levels of dopaminergic metabolites appear to reflect an increase in the enzymatic degradation of dopamine, whose levels increased considerably after the administration of the psychoactive substances. In these experimental conditions, the enzymes involved in dopamine degradation would multiply their efforts to neutralize it and thus avoid the oxidative stress caused by the oxidation of this neurotransmitter. However, the increase in dopamine levels and its metabolites was not proportional, which could be because the processes of reuptake and metabolization of dopamine do not occur at the same rate as its release.

In summary, the different substances studied exerted a considerable stimulating effect on the brain’s reward system, and when administered in combination, additive and synergistic effects on dopamine levels were observed. These results may have important practical implications since, although natural rewards also lead to an increase in dopamine release (especially in the NAcc), the effect is not as robust or prolonged as that caused by most psychoactive substances [61]. Our results suggest that repeated consumption of mixtures of caffeine, nicotine, and/or ethanol could raise reward thresholds for both natural and drug-related rewards, potentially leading to an escalation in the consumption of rewards to reach that threshold. Additionally, it should not be overlooked that nerve cells lack sufficient mechanisms to cope with excessively high and persistent levels of dopamine induced by psychoactive substances, which may contribute to the development of oxidative stress and consequent irreversible damage to the nervous system.

## 5. Conclusions

In the present study, it has been observed that caffeine, nicotine, and ethanol exerted a stimulating effect on both the nigrostriatal dopaminergic pathway, implicated in drug-seeking and consumption behaviors, and the mesocorticolimbic pathway, associated with reward, motivation, memory, and cognition. Overall, when these substances were administered in combination, they produced additive, synergistic, and potentiating effects, increasing each other’s neurochemical effects, or at least the substance with the predominant effect on reward-related dopaminergic circuits maintained its impact. The increased activation of the reward circuitry when consuming various psychoactive substances together may contribute to the high incidence of co-abuse, promoting both motivation and behaviors related to the consumption of other psychoactive substances.

## Figures and Tables

**Figure 1 biomedicines-12-02591-f001:**
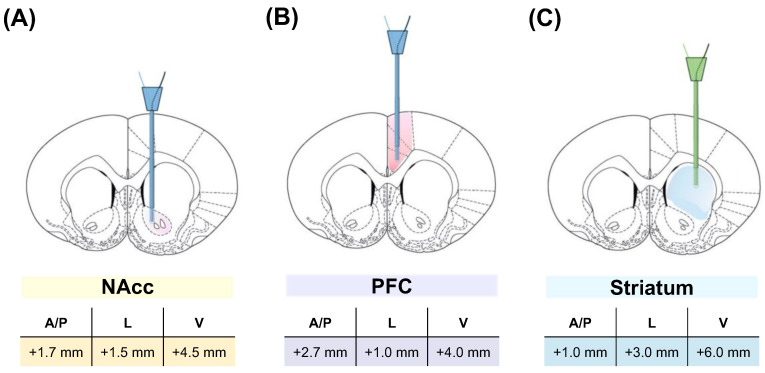
Schematic diagram of the location and coordinates (according to Paxinos and Watson, 1998) of microdialysis probes in the NAcc (**A**), PFC (**B**), or striatum (**C**).

**Figure 2 biomedicines-12-02591-f002:**
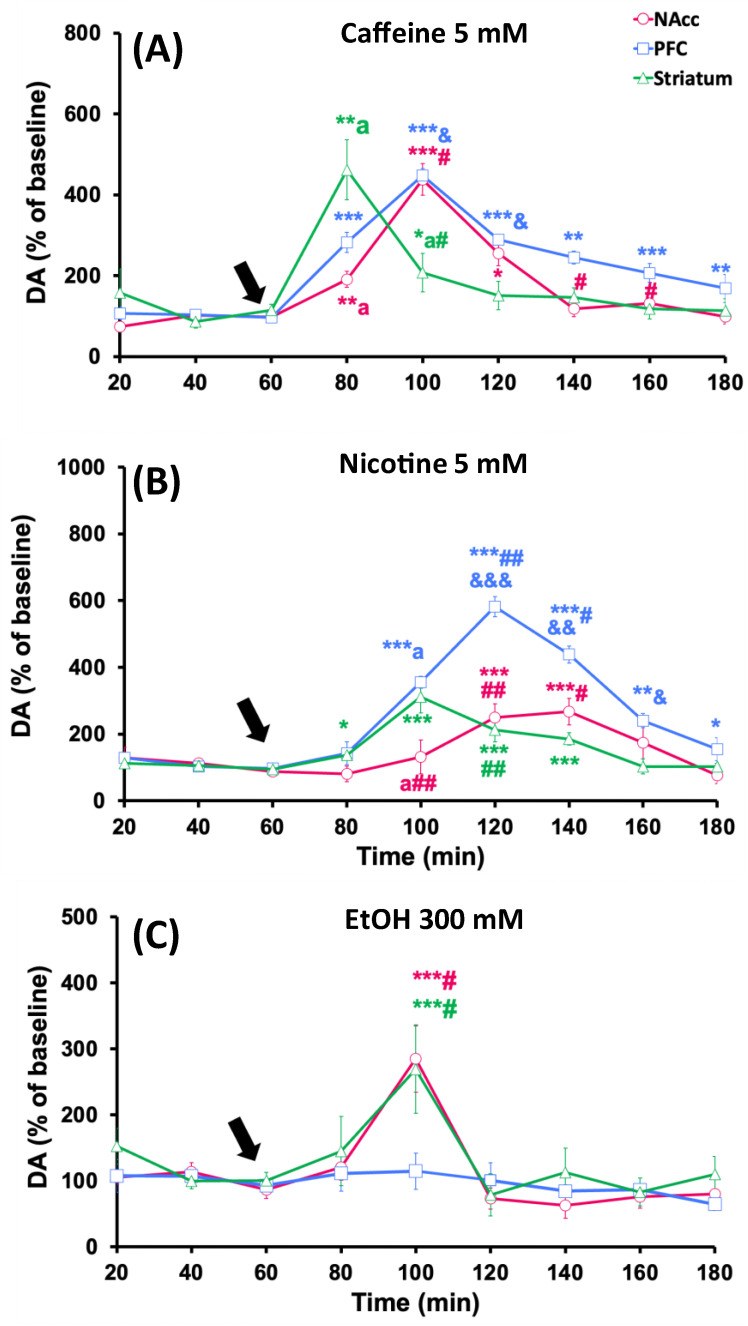
Effect of individual administration of caffeine (**A**), nicotine (**B**), and ethanol (**C**) on dopamine levels in the NAcc, PFC, and striatum of rats. Substances were perfused for 60 min starting from the time indicated by the arrow. Results are shown as mean ± S.E.M., expressed as a percentage of baseline levels (100%). Baseline levels were considered as the average of dopamine concentrations in the two samples collected before treatment administration. Significant differences: * *p* < 0.05, ** *p* < 0.01 and *** *p* < 0.001 with respect to basal levels; ^a^
*p* < 0.05 comparing NAcc vs. striatum; ^#^
*p* < 0.05 and ^##^
*p* < 0.01 comparing NAcc vs. PFC; ^&^
*p* < 0.05, ^&&^
*p* < 0.01 and ^&&&^
*p* < 0.001 comparing PFC vs. striatum.

**Figure 3 biomedicines-12-02591-f003:**
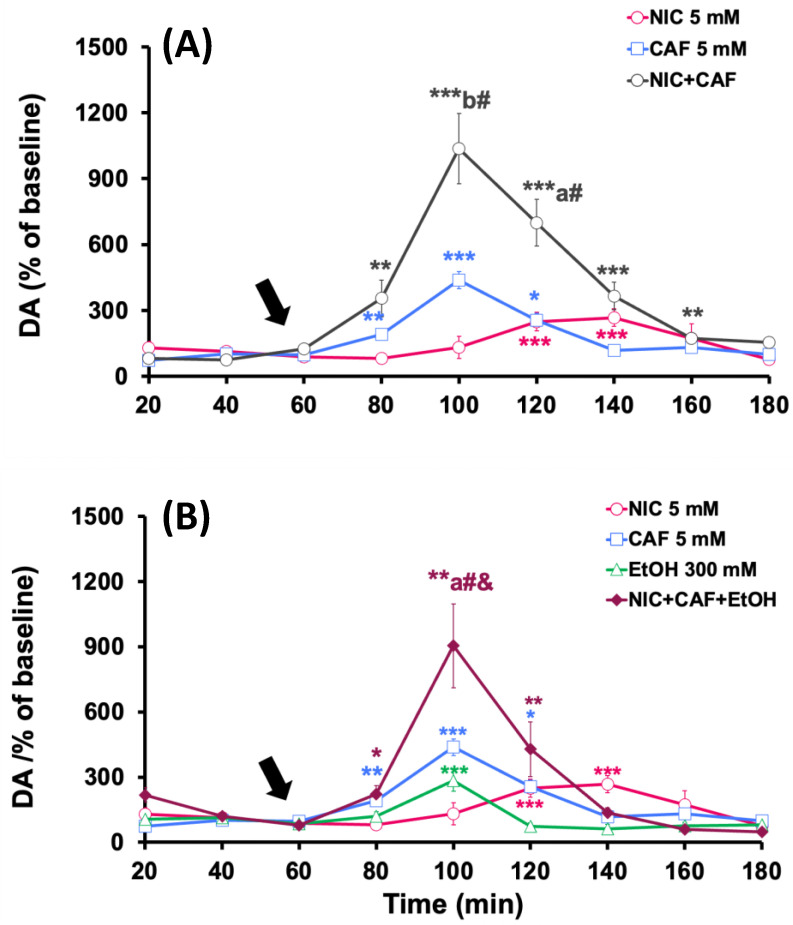
Effect of individual and combined administration of (**A**) nicotine (NIC) + caffeine (CAF) and (**B**) NIC + CAF + ethanol (EtOH) on dopamine levels in the NAcc. Treatments were administered starting from the time indicated by the arrow. Results are shown as mean ± S.E.M., expressed as a percentage of baseline levels (100%). Baseline levels were considered as the average of dopamine concentrations in the two samples collected before treatment administration. Significant differences: * *p* < 0.05, ** *p* < 0.01 and *** *p* < 0.001 with respect to basal levels; ^a^
*p* < 0.05 and ^b^
*p* < 0.01 with respect to nicotine group; ^#^
*p* < 0.05 with respect to caffeine group; ^&^
*p* < 0.05 with respect to ethanol group.

**Figure 4 biomedicines-12-02591-f004:**
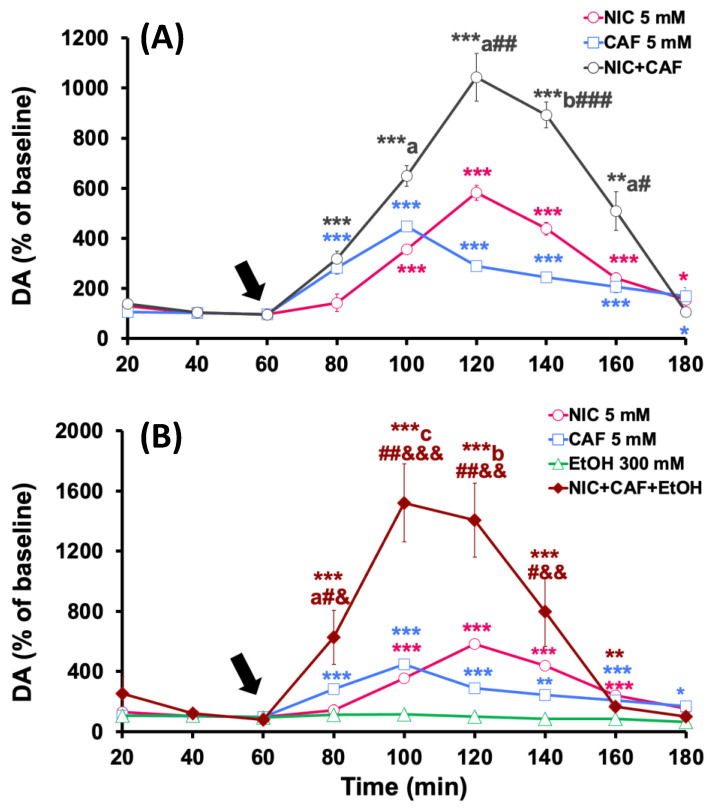
Effect of individual and combined administration of (**A**) nicotine (NIC) + caffeine (CAF) and (**B**) NIC + CAF + ethanol (EtOH) on dopamine levels in the PFC. Treatments were administered starting from the time indicated by the arrow. Results are shown as mean ± S.E.M., expressed as a percentage of baseline levels (100%). Baseline levels were considered as the average of dopamine concentrations in the two samples collected before treatment administration. Significant differences: * *p* < 0.05, ** *p* < 0.01 and *** *p* < 0.001 with respect to basal levels; ^a^
*p* < 0.05, ^b^
*p* < 0.01 and ^c^
*p* < 0.001 with respect to nicotine group; ^#^
*p* < 0.05, ^##^
*p* < 0.01 and ^###^
*p* < 0.001 with respect to caffeine group; ^&^
*p* < 0.05, ^&&^
*p* < 0.01 and ^&&&^
*p* < 0.001 with respect to ethanol group.

**Figure 5 biomedicines-12-02591-f005:**
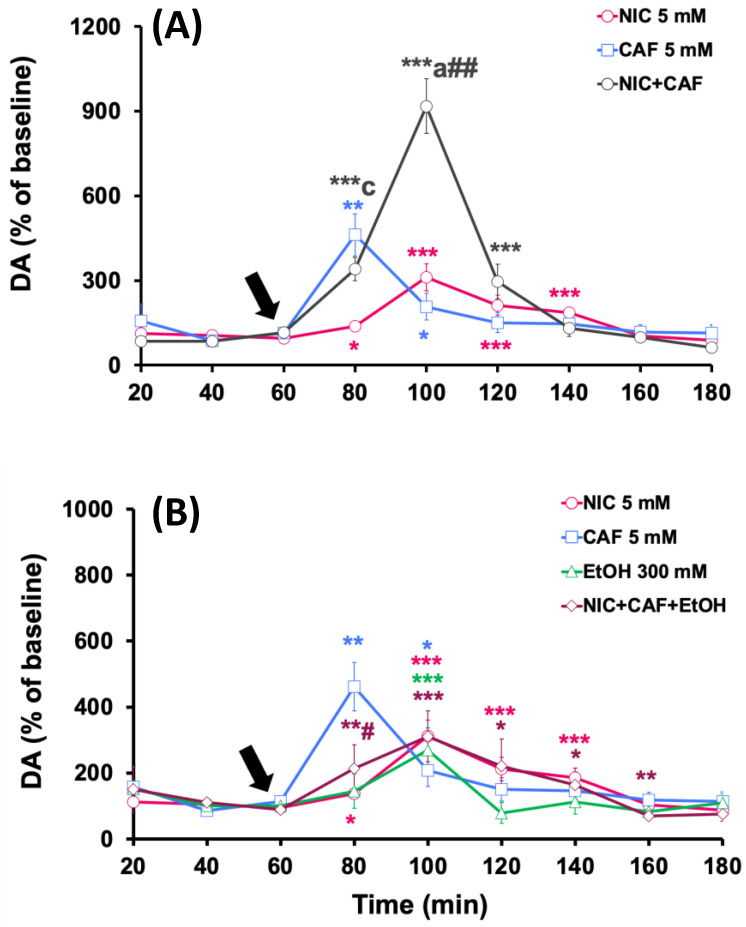
Effect of individual and combined administration of (**A**) nicotine (NIC) + caffeine (CAF) and (**B**) NIC + CAF + ethanol (EtOH) on dopamine levels in the striatum. Treatments were administered starting from the time indicated by the arrow. Results are shown as mean ± S.E.M., expressed as a percentage of baseline levels (100%). Baseline levels were considered as the average of dopamine concentrations in the two samples collected before treatment administration. Significant differences: * *p* < 0.05, ** *p* < 0.01 and *** *p* < 0.001 with respect to basal levels; ^a^
*p* < 0.05 and ^c^
*p* < 0.001 with respect to nicotine group; ^#^
*p* < 0.05 and ^##^
*p* < 0.01 with respect to caffeine group.

**Figure 6 biomedicines-12-02591-f006:**
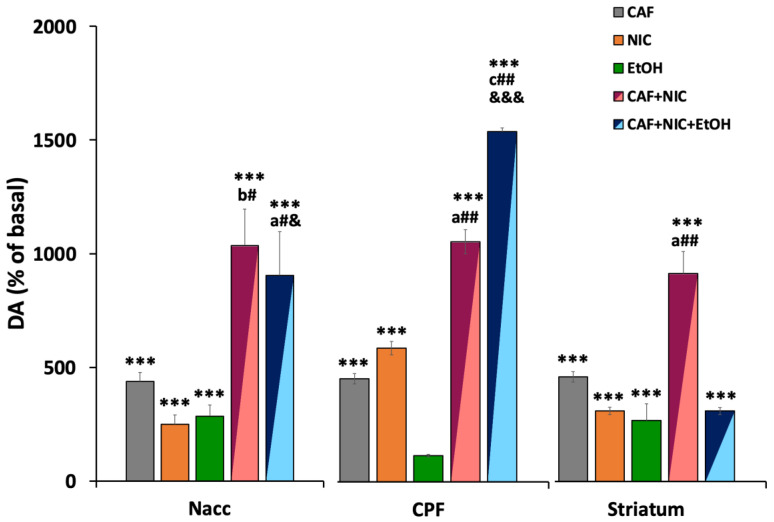
Comparative effects of nicotine (NIC, 5 mM), caffeine (CAF, 5 mM), ethanol (EtOH, 300 mM), and the combinations NIC + CAF and NIC + CAF + EtOH on dopamine levels in the NAcc, PFC, and striatum. The graphs show the synergistic effect of the administration of caffeine together with nicotine in the NAcc and striatum and that the addition of ethanol to this mixture produces an additive effect on the release of dopamine. The values observed are the maximal increases observed in Figure 2, Figure 3, Figure 4 and Figure 5, expressed as a percentage of the basal levels (100%). Significant differences: *** *p* < 0.001 with respect to basal levels; ^a^
*p* < 0.05, ^b^
*p* < 0.01 and ^c^
*p* < 0.001 with respect to nicotine group; ^#^
*p* < 0.05 and ^##^
*p* < 0.01 with respect to caffeine group; ^&^
*p* < 0.05 and ^&&&^
*p* < 0.001 with respect to ethanol group.

**Figure 7 biomedicines-12-02591-f007:**
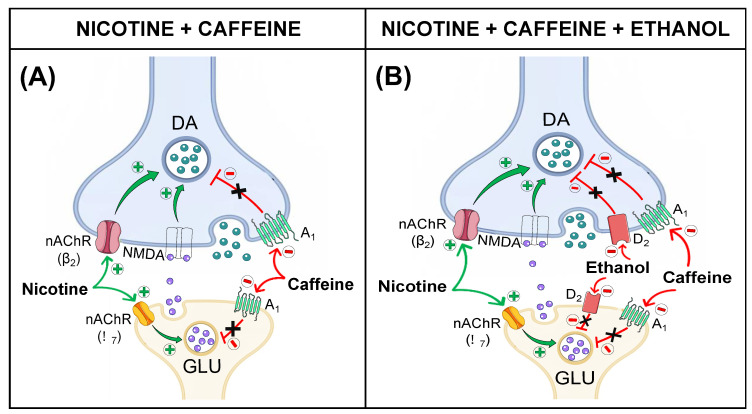
Schematic representation of potential mechanisms underlying the effects of combined administration of (**A**) nicotine and caffeine and (**B**) nicotine, caffeine, and ethanol. Abbreviations: DA, dopamine; nAChR, nicotinic acetylcholine receptor; NMDA, N-methyl-D-aspartate receptor; A1, adenosine A1 receptor; GLU, glutamate; D2, dopamine D2 receptor.

**Table 1 biomedicines-12-02591-t001:** General design of microdialysis experiments. Ringer solution was perfused for the first 60 min, followed by administration of the corresponding treatment (caffeine (CAF), nicotine (NIC), ethanol (EtOH), or their combinations), and finally, Ringer solution was administered again during the last hour of the experiment.

Group/Area		Treatment Protocol 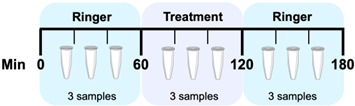			
		*n*	60 min	60 min	60 min
CAF 5 mM	NAcc	5	Ringer	CAF	Ringer
	PFC	6			
	Striatum	4			
NIC 5 mM	NAcc	5	Ringer	NIC	Ringer
	PFC	5			
	Striatum	5			
EtOH 300 mM	NAcc	6	Ringer	EtOH	Ringer
	PFC	4			
	Striatum	6			
NIC 5 mM + CAF 5 mM	NAcc	5	Ringer	NIC + CAF	Ringer
	PFC	5			
	Striatum	6			
NIC 5 mM + CAF 5 mM + EtOH 300 mM	NAcc	6	Ringer	NIC + CAF + EtOH	Ringer
	PFC	6			
	Striatum	7			

**Table 2 biomedicines-12-02591-t002:** Maximal effects of caffeine (CAF), nicotine (NIC), ethanol (EtOH), and the combinations NIC + CAF and NIC + CAF + EtOH on DOPAC and HVA levels in the NAcc, PFC, and striatum of rats. Values are represented as mean ± S.E.M., expressed as a percentage change from baseline (100%), calculated as the average of the two values before treatment administration. Significance level: * *p* < 0.05, ** *p* < 0.01 and *** *p* < 0.001 with respect to basal levels; ^&^
*p* < 0.05 with respect to ethanol group.

Group/Treatment	DOPAC		
	NAcc	PFC	Striatum
Basal	102.2 ± 10.7	103.5 ± 11.2	102.2 ± 8.6
CAF 5 mM	122.2 ± 5.1 **	194.4 ±37.7 **	129.6 ± 6 **
NIC 5 mM	117.6 19.7	280.3 ± 38 ***	117.6 ± 19.7
CAF + NIC	134.5 ± 26.4	191.2 ± 27.2 **	158.9 ± 32
EtOH 300 mM	163.2 ± 40.4 *	71.9 ± 12.2	203.1 ± 35.3 *
CAF + NIC + EtOH	146.7 ± 20.0 ***	301 ± 38.6 ***^&^	145.6 ± 13.5 ***
	**HVA**		
Basal	104.7 ± 6.8	116.1 ± 17.2	109.6 ± 9.3
CAF 5 mM	107.8 ± 16.3	134.8 ± 13.3	128 ± 9.6 *
NIC 5 mM	137.8 ± 22.8 *	129 ± 18.1	146.9 ± 22.3 *
CAF + NIC	139.7 ± 22.9 *	146.3 ± 29.8	134.8 ± 18.7
EtOH 300 mM	116.6 ± 17.5	81.3 ± 16.1	131.6 ± 9.3 *
CAF + NIC + EtOH	160.5 ± 23.1 ***	168.4 ± 27.6 *	119.6 ± 15.8

## Data Availability

No data were used for the research described in the article.

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
