# Peer review of "Possible Potentiating Effects of Combined Administration of Alcohol, Caffeine, and Nicotine on In Vivo Dopamine Release in Addiction-Related Circuits Within the CNS of Rats"

_biomedicines, 2024, doi:10.3390/biomedicines12112591_

Round 1

Reviewer 1 Report

Comments and Suggestions for Authors

The manuscript shows important effects of three drugs widely used around the world and  brings several resultst would be interesting.  In addition to describing what was observed experimentally. I have two questions :

1. Is calcium necessary for dopamine  release?

2. dopamine release is an direct  or indirect-acting of these three  drugs, in other words,  alcohol, caffeine and nicotine enhances the release or action of  endogenous neurotransmitter, such as L~glutamate , but has no specific effect on the release of dopamine. 

Author Response

Comment 1: Is calcium necessary for dopamine  release?

First of all, we are very grateful for the comments of the reviewer to improve our manuscript.

Regarding the first issue, the findings from the available studies suggest that the dopamine release induced by each of the three substances examined (caffeine, nicotine, and ethanol) is indeed dependent on the presence of calcium in the cellular cytosol. In this way, the presence of calcium in the intracellular environment of dopaminergic neurons facilitates the exocytotic release of dopamine into the synaptic cleft. However, how each of these substances affects intracellular calcium concentrations is complex and varies significantly.

On one hand, caffeine is thought to promote an increase in intracellular calcium levels by inducing its release from intracellular stores. Specifically, caffeine activates ryanodine receptors in the endoplasmic reticulum, leading to the subsequent efflux of calcium into the cellular cytosol, thereby facilitating the release of dopamine into the synaptic cleft.

On the other hand, nicotine primarily acts on nicotinic acetylcholine receptors (nAChRs), the activation of which allows the influx of cations, including calcium ions, into the neuron. This results in membrane depolarization and the subsequent opening of voltage-gated calcium channels (especially N-type and P/Q-type channels), which permit the entry of calcium into the intracellular environment and the subsequent release of dopamine.

Finally, the dopamine release induced by ethanol also significantly depends on calcium influx into neurons, although the specific mechanism may be more complex and vary depending on the type of neuron and the brain region involved. In general, it is thought that ethanol promotes membrane depolarization and the subsequent opening of voltage-gated calcium channels, allowing calcium to enter the neuronal cytosol. Additionally, ethanol may also influence intracellular signaling pathways that modulate the release of calcium from intracellular stores.

Therefore, in accordance with the reviewer's comment, it appears that calcium may play a significant role in the changes observed in our study. However, this article does not aim to identify all the specific mechanisms by which caffeine, nicotine, and/or ethanol induce changes in dopamine release. This is why in the discussion of this manuscript we have only mentioned the main molecular targets of each of the substances under study. That said, it would be highly interesting for future research to focus on characterizing the specific mechanisms underlying the dopamine release induced by the combination of the substances under investigation, particularly the potential mediating role of calcium in these effects.

Comment 2: Dopamine release is a direct or indirect-acting of these three drugs, in other words,  alcohol, caffeine and nicotine enhances the release or action of endogenous neurotransmitters, such as L~glutamate, but has no specific effect on the release of dopamine. 

In the discussion of this manuscript, we have hypothesized the different mechanisms that could underlie the effects observed in our study based on the available literature. In general, it is proposed that the dopamine release described would be the result of both direct and indirect actions of each of the three drugs on the dopaminergic system.

On the one hand, the effects of caffeine, nicotine, and alcohol appear to be mediated by A1 adenosine receptors (A1Rs), nicotinic acetylcholine receptors (nAChRs), and D2 dopamine receptors located in dopaminergic terminals, respectively (direct action). Nonetheless, caffeine and nicotine could also act on A1Rs and nAChRs located in glutamatergic terminals in the studied regions. In this way, the specific action of each of these substances on glutamatergic terminals leads to an increase in glutamate release, which in turn enhances dopamine release by activating ionotropic glutamate receptors located in dopaminergic terminals (indirect action).

Therefore, as we have attempted to reflect in the discussion of this work, the available mechanistic studies on caffeine, nicotine, and alcohol suggest that their effects on the dopaminergic system involve both direct and indirect mechanisms of action.

Reviewer 2 Report

Comments and Suggestions for Authors

The aim of the present experiments was to assess the dopamine level in the nucleus accumbens (NAcc), the prefrontal cortex (PFC) and the striatum – bran structures related to drug-seeking and consumption behaviors. The results showed the stimulating effects of the examined psychoactive substances on the reward pathways in the central nervous system (CNS). Moreover, the results can be of relevance for the comprehension of the neurochemical reaction provoked by the interactions between these substances in the CNS. Overall this is a well-designed study and provides very interesting results.

The authors have taken up an important topic - determining whether the combined use of these three substances causes an additive effect or a synergistic effect. The choice of these three substances is a very good choice, because they are popular psychoactive substances, widely available and often used and abused, and also often used together. The authors have shown the expected synergistic effects of these substances on the level of dopamine in brain structures related to addiction. A very strong point of the work is the conclusion that such a synergistic effect resulting from the use of these substances together may predispose to a more severe addiction and may promote both motivation and behaviors related to the consumption of other psychoactive substances.

The work is very carefully researched and written in clear, accessible language.

This paper is deserving of publication.

Author Response

We greatly appreciate the effort made during the review of this article, as well as the feedback provided by the reviewer.